# Is It Advisable to Use Probiotics Routinely After a Colonoscopy? A Rapid Comprehensive Review of the Evidence

**DOI:** 10.3390/medsci13020076

**Published:** 2025-06-09

**Authors:** Matteo Piciucchi, Alice Rossi, Alissa Satriano, Raffaele Manta

**Affiliations:** 1Gastroenterology and Endoscopy South Area Unit, Medical Department, “Santa Maria della Stella” Hospital of Orvieto USL Umbria2, 05018 Orvieto, TR, Italy; 2Clinical Nutrition and Dietetics Unit, “Santa Maria della Misercordia” University Hospital of Perugia, 06122 Perugia, PG, Italy; alissa.satriano@ospedale.perugia.it; 3Gastroenterology and Digestive Endoscopy Unit, ASL Toscana Nord-Ovest, “Riuniti” Hospital, 56121 Livorno, LI, Italy; raffaelemanta4@gmail.com

**Keywords:** post-colonoscopy symptoms, dysbiosis, probiotics

## Abstract

About 5–20% of patients who undergo colonoscopy, in the days and weeks following the procedure, develop various symptoms (abdominal pain, bloating, and bowel alteration) mainly related to dysbiosis induced by the propaedeutic intestinal preparation. Some studies have positively evaluated the impact of the administration of different mixtures of probiotics in preventing and/or limiting this symptomatology. The aim of this review is to evaluate and summarize the available scientific evidence supporting the use of probiotics post-colonoscopy and to define their real efficacy as a routine treatment in a clinical setting.

## 1. Introduction and Aim

It has been observed that between 5 and 20% of all patients undergoing a diagnostic colonoscopy experience a variety of post-procedural symptoms characterized by changes in bowel habits (both constipation and diarrhea), flatulence, and abdominal discomfort/pain, which can persist in the days and weeks following the examination, negatively impacting the patient’s daily activities [1,2].

The onset of this sort of “post colonoscopy syndrome” is thought to be linked to two types of etiopathogenic mechanisms:(a)Direct mechanical factors related to trauma caused by the endoscopic instrument during exploration: non-transmural wall damage related to the passage of angles, mechanical stretching of the mesenteries and/or intestinal adhesions, non-transmural thermal damage during interventional procedures, etc.(b)Indirect factors related to the alteration of the intestinal microbiota mainly resulting from the intake of the different preparatory preparations used for the exam (single or split dose; high or low volume preparation) [1,2,3].

With specific regard to indirect factors, it has been observed that intestinal preparation, especially when using a high-volume single-dose approach, significantly decreases and alters the patient’s normal bacterial flora [1,2,3,4].

Several studies on healthy volunteers have observed a general decrease in anaerobic bacteria after a colonoscopy (Bacilli, including various strains of *Lactobacillus*) with a peculiar depletion of *Firmicutes* and *Bacteroidetes* strains and a parallel increased amount of aerobic *Proteobacteria*. These alterations could be potentially related both to the over-intake of oxygen during the introduction of the oral purgative (especially in large amounts of a high-volume formulation) and to the insufflation of O2 during the examination, which are factors that subvert the conditions of physiological intestinal bacterial homeostasis [1,2,3,4].

In particular, in a study conducted by Drago et al. in 2015 on the stool of patients undergoing colonoscopy, it was observed that the stool samples of patients who had taken the high-volume preparation showed a persistent reduction in the concentrations of *Enterobacteriaceae* and *Lactobacillaceae* up to 30 days after taking the high-volume purgative [3].

Although dysbiosis-related post-colonoscopy symptoms are a minor and self-limiting complication of diagnostic endoscopy of the lower digestive tract, the duration and severity of the symptoms can negatively impact the patient’s quality of life and lead to loss of working days (with significant socioeconomic impact) in the days and weeks following the procedure [1].

This is where the rationale for using probiotic mixtures to prevent and treat post-colonoscopy symptoms related to intestinal dysbiosis comes from.

Since this topic has been seldom addressed in the literature, the aim of this rapid communication review is to evaluate and summarize the available scientific evidence supporting the use of probiotics post-colonoscopy and to define their real efficacy as a routine treatment in a clinical setting.

## 2. Methods

Articles were retrieved from the MEDLINE, EBSCO-Host, Cochrane, ProQuest, SCOPUS, and Google Scholar electronic databases, utilizing specific keywords (“Probiotic” OR “Probiotics”) AND (“Colonoscopy” OR “Colonoscopies”).

## 3. Results

To date, to our knowledge, only five randomized controlled probiotic/placebo clinical studies (three of which were double-blind, high-quality randomized controlled trials; see Table 1) and three non-randomized studies have evaluated the post-colonoscopy use of different probiotic mixtures in patients without specific pre-existing conditions (Table 1).

All of these studies have demonstrated, to different degrees, the positive impact of probiotic treatments in preventing or limiting post-colonoscopy symptoms (a reduction in abdominal pain intensity and bloating in particular) and/or dysbiosis [5,6,7,8,9,10,11,12].

**Table 1 medsci-13-00076-t001:** Clinical studies evaluating probiotics administration after a colonoscopy.

Study	Patients	Probiotics Mixture and Formulation	Duration of Treatment	Outcome
D’Souza B et al. [8] Double-blind RCT High quality	133 probiotics vs. 126 controls	*Lactobacillus acidophilus* NCFM and *Bifidobacterium lactis* Bi-07 in capsule formulation for both the probiotic and the placebo.	Once daily for 14 days after the colonscopy.	↓ Number of pain days, especially in patients with pre-existing symptoms. (After 21 days of clinical evaluation.)
Labenz J et al. [7] Double-blind RCT High quality	36 probiotics vs. 35 controls	*Bifidobacterium bifidum* W23, *Bifidobacterium lactis* W51, *Enterococcus faecium* W54, *Lactobacillus acidophilus* W37, *Lactobacillus rhamnosus* WGG, and *Lactococcus lactis* W19 in powder formulation for both the probiotic and the placebo.	Twice daily for 30 days after the colonscopy.	↓ Constipation days. (After 28 days of clinical evaluation.)
Hung J-S [12] Single-blind RCT Low/moderate quality (lack of blinding/poor objective assessment of outcome)	71 controls vs. 72 probiotics (7 days before/after colonoscopy) 67 probiotics (7 days before colonoscopy) 74 probiotics (7 days after colonoscopy)	*Lactobacillus casei* sp. *rhamnosus* GG (LGG) in capsule formulation.	Thrice daily 7 days before/after the colonoscopy. Thrice daily 7 days before the colonoscopy. Thrice daily 7 days after the colonoscopy.	↓ Abdominal pain intesity. ↓ Indigestion symptoms. (In all probiotics groups after 7 days of clinical evaluation.)
Bonavina L et al. [9] Single-arm clinical survey Low quality (single arm/lack of blinding/poor objective assessment of outcome)	2.979 probiotics (no control group)	*Lactobacillus plantarum* LP01, *Lactobacillus lactis subspecies cremoris* LLC02, and *Lactobacillus delbrueckii* LDD01 in powder formulation.	Once daily for 28 days after the colonscopy.	↓ Abdominal pain intensity. ↓ Bloating. (After 28 days of clinical evaluation.)
Aragona SE et al. [10] Open RCT Low/moderate quality (lack of blinding/poor objective assessment of outcome)	3.197 probiotics vs. 1.612 controls	*Lactobacillus plantarum* LP01, *Lactobacillus lactis subspecies cremoris* LLC02, and *Lactobacillus delbrueckii* LDD01 in powder formulation for both the probiotic and the placebo.	Once daily for 28 days after the colonscopy.	↓ Abdominal pain intensity. ↓ Bloating. ↓ Abdominal discomfort. ↓ Bowel alteration. (After 28 days of clinical evaluation.)
Mullaney T et al. [6] Double-blind RCT High quality	75 probiotics vs. 75 controls	*Lactobacillus acidophilus* NCFM and *Bifidobacterium lactis* Bi-07 in capsule formulation for both the probiotic and the placebo.	Once daily for 14 days after the colonscopy.	↓ Bloating in patients with pre-existing symptoms. (After 14 days of clinical evaluation.)
Deng X et al. [5] Fecal sample analysis Low quality (lack of blinding/low sample analyzed/poor objective assessment of outcome)	16 probiotics vs. 16 controls	*Bifidobacterium infantis*, *Lactobacillus acidophilus*, *Enterococcus faecalis*, and *Bacillus cereus* in tablet formulation for both the probiotic and the placebo.	Thrice daily for 5–7 days after the colonoscopy.	Improvement of intestinal dysbiosis. (After 7 days of stool evaluation.)
Chen Z et al. [11] Fecal sample analysis Low quality (lack of blinding/low sample analyzed/poor objective assessment of outcome)	7 probiotics vs. 4 controls	*Clostridium butyricum* in capsule formulation.	Thrice daily for 20 days after the colonoscopy.	Improvement of intestinal dysbiosis (After 7–30–60 days of stool evaluation.)

RCT: Randomized controlled trial. ↓: decrease.

The probiotics strains used to date in the various studies with positive results have been the following:(a)A mixture of *Bifidobacterium bifidum* W23, *Bifidobacterium lactis* W51, *Enterococcus faecium* W54, *Lactobacillus acidophilus* W37, *Lactobacillus rhamnosus* WGG, and *Lactococcus lactis* W19 [7].(b)A mixture of *Lactobacillus acidophilus* NCFM and *Bifidobacterium lactis* Bi-07 [6,8].(c)A mixture of *Lactobacillus plantarum* LP01, *Lactobacillus lactis subspecies cremoris* LLC02, and *Lactobacillus delbrueckii* LDD01 [9,10].(d)A mixture of *Bifidobacterium infantis*, *Lactobacillus acidophilus*, *Enterococcus faecalis*, and *Bacillus cereus* [5].(e)*Clostridium butyricum* [11].(f)*Lactobacillus casei* sp. *rhamnosus* GG [12].

In particular, all three published double-blind randomized controlled trial studies used strains of *lactobacilli* in combination with *bifidobacterium* and highlighted a statistically significant improvement of gastrointestinal symptoms after endoscopic examination compared to the placebo [6,7,8].

Respectively in these three studies, a reduction in the number of days of pain [8], constipation [7], and bloating [6] was observed.

The indication for colonoscopy does not seem to influence the response to treatment, but, in two randomized controlled trials, the administration of probiotics resulted in greater efficacy in the subgroup of patients who had pre-existing symptoms before the endoscopic examinations were performed [6,8].

## 4. Discussion

The encouraging results of these studies therefore may suggest that the use of probiotics could become a routine aid in patients undergoing diagnostic colonoscopy to prevent and/or reduce post-procedural symptoms related to dysbiosis.

However, to date, the studies in the literature, which are few and heterogeneous (especially regarding the differing mixtures of the administered probiotics), do not yet allow us to draw definitive conclusions on the efficacy of routine treatment with probiotics post-colonoscopy: the results of a recent meta-analysis that attempted to summarize the data, in fact, do not highlight a clear therapeutic gain in patients who have been given probiotic treatment post-colonoscopy [13].

In fact, many gray areas still need to be clarified regarding the characteristics of the symptoms and the optimal therapy: Which type of bowel preparation has the greatest impact on the onset of symptoms and which categories of patients are most at risk? Which strains of probiotics are best for resolving symptoms, at what dosage, and for how long?

Regarding this last point in particular, we have previously described how the probiotics predominantly used for therapy in the published studies have been mixtures of *lactobacillus* and *bifidobacterium*.

Although the mechanisms of actions of these two strains have not yet been entirely comprehended, Deng et al. [5] observed that probiotics containing primarily *Lactobacillus* and *Bifidobacterium* could mitigate the negative effect of the over-intake of O2 from bowel preparation on anaerobic intestinal bacteria (a decrease in *Bacteroidetes* and *Firmicutes* and an increase in *Proteobacteria*), as they greatly increase the abundance of *Bacteroidetes*. Butyrate produced by *Bacteroidetes* in fact plays an important role in maintaining the intestinal health of the host by regulating and ensuring immunity and applying an antitumor effect [14].

However, these bacterial strains may not necessarily be the best probiotics for treatment, which is a legitimate doubt that arises from the analysis of the physiopathology of the damage induced by the purgative preparation on the colon microbiome.

In fact, the effect of intestinal preparation for a colonoscopy (especially when using a high-volume single-dose approach) on the individual’s microbiome, as previously mentioned, greatly increases the concentration of O2 in the colonic environment, and interestingly it has been recently hypothesized that it may determine a sort of regression of the microbiome to the neonatal pediatric bacterial colonization phase. Unlike adults, in the neonatal phase, the individual’s microbiome in fact predominantly presents with the development of aerobes bacteria, which, following the consumption of O2 in the intestinal lumen, determines the subsequent development of anaerobic or facultative anaerobic strains such as *Enterobacteriaceae* [15].

On the basis of this hypothesis, it might seem conceivable that probiotic strains of O2 consumers extracted from the feces of healthy newborns could constitute a valid aid for a physiological bacterial recolonization of the colon after the intake of a purgative for colonoscopy. Therefore, in purely theoretical terms, we hypothesize that the administration of non-pathogenic neonatal strains of *Enterobacteriaceae* such as *Escherichia Coli* (which, as we previously mentioned, are markedly reduced in the long term after the intake of the preparation) could constitute an adequate treatment for post-colonoscopy syndrome, but clinical studies are needed to confirm this supposition [3,4,15].

In conclusion, more prospective multi-arm case-control studies on large case series are certainly needed to establish the real efficacy and necessity of probiotic treatments after colonoscopy. There is a wide variability of proposed treatments that have not been compared with each other and no cost-effectiveness analysis is yet available in the literature. Therefore, we are still far from being able to suggest a routine probiotics treatment after colonoscopy.

## Data Availability

No new data were created.

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
