# Peer review of "Is It Advisable to Use Probiotics Routinely After a Colonoscopy? A Rapid Comprehensive Review of the Evidence"

_medsci, 2025, doi:10.3390/medsci13020076_

Round 1
Reviewer 1 Report
Comments and Suggestions for Authors
The authors clearly and simply present the beneficial effects of probiotic treatment on the side effects caused by colonoscopy, demonstrating the usefulness of probiotics in improving the quality of life of patients undergoing this procedure.
Although the document is clear and simple, I consider it necessary to deepen and clarify the information presented in the table (days of treatment, days of evaluations, duration of beneficial effects, etc.).
Regarding the format, it is necessary to:
1. Review the abbreviations and their meanings (e.g., RCT, ptx).
2. The name of microorganisms is generally written in italics.
3. The reference format is heterogeneous; review the journal's guidelines.
Author Response
Dear reviewer,
I believe that your suggestions have been valuable in improving the quality of my work.
I modified the work according to what you indicated to me in the following way:
- I added a new column to the table (entitled “duration of treatment”) in which I included the treatment days and the mode of probiotic intake. In the last column of the table (entitled “outcome”), I have included the number of days of clinical assessment for each study. Unfortunately, no study has assessed how lasting the benefit of the treatment has been over time, so I could not add this data, as you requested.
- I have included a legend for the abbreviations (RCT, pts) at the bottom of the table.
- I wrote the names of the microorganisms in italics both in the text and on the table.
- I standardized the format of the references according to the journal's guidelines.
Thank you very much for taking the time to review this manuscript.
Best regards,
Dr Matteo Piciucchi MD, PhD
Reviewer 2 Report
Comments and Suggestions for Authors
The studies that are used for this review are extremely heterogeneous. Some have no control groups and different outcomes are measured - symptoms vs. gut flora.
Without seeing the results of these studies in more detail, the reader cannot draw any conclusions about the validity of these studies and how the might impact clinical practice.
Does indication for scope affect response?
What are the reductions in pain/bloating?
What bowel prep was associated with better/worse outcomes?
Author Response
Dear reviewer,
I believe that your suggestions have been valuable in improving the quality of my work.
I modified the work according to what you indicated to me in the following way:
Comment 1: Does indication for scope affect response?
Response 1: To respond to your correct observation, I have modified the text from line 103 to line 106 as follows: “The indication for colonoscopy does not seem to influence the response to treatment, but in two randomized controlled trial, the administration of probiotics results to be more efficacy in subgroup of patients with preexisting symptoms before endoscopic examinations.”
Comment 2: What are the reductions in pain/bloating?
Response 2: Compared to the original version, I better specified in the 'outcome' column of the table whether the probiotic was associated with a reduction in the number of days associated with pain, constipation, or bloating, or whether the probiotic resulted in a reduction of the intensity of abdominal pain. Unfortunately, the studies present in the literature do not include additional elements (such as a VAS scale for pain) to better clarify this point.
Comment 3: What bowel prep was associated with better/worse outcomes
Response 3: In the section of the text that goes from line 50 to line 52, I have better specified what you requested in the following way: “With specific regard to indirect factors, it has been observed that intestinal preparation, especially in the high-volume one-dose mode, significantly decreases and al-ters the patient's normal bacterial flora.” I further emphasized this concept in lines 56-59 of text: “These alterations could be potentially related both to the oxygen over intake during the introduction of the oral purgative (especially in large amount high-volume formula-tion) and to the insufflation of O2 during the examination, factors that subvert the conditions of physiological intestinal bacterial homeostasis” and in lines 134-136 “In fact, the effect of intestinal preparation (especially in the high-volume one-dose mode) for colonoscopy on the individual's microbiome, as previously mentioned, greatly increases the concentration of O2 in the colonic environment”.
Thank you very much for taking the time to review this manuscript.
Best regards,
Dr Matteo Piciucchi MD, PhD
Reviewer 3 Report
Comments and Suggestions for Authors
This paper addresses a clinically relevant question regarding probiotic use post colonoscopy. While the topic is of importance, major revisions are needed. Here are my concerns-
(1) clarifying the search strategy to enhance reproducibility, including quality assessment of included RCTs to strengthen methodological rigor.
(2) re-analyze the included RCTs rather than briefly telling the results from each trial.
(3) discussion would benefit from mechanistic insights into strain-specific effects.
Author Response
Dear reviewer,
I believe that your suggestions have been valuable in improving the quality of my work.
I modified the work according to what you indicated to me in the following way:
Comment 1: Clarifying the search strategy to enhance reproducibility, including quality assessment of included RCTs to strengthen methodological rigor.
Response 1: To respond to your correct observation, I modified the text between lines 72-78 as follows: “To date, to our knowledge, 5 randomized controlled probiotic/placebo clinical studies (3 of which were double-blind high quality RCT, see Table 1) and 3 non-randomized studies have evaluated the post-colonoscopy use of different probi-otic mixtures in patients without specific pre-existing conditions (table 1). Articles were retrieved from MEDLINE, EBSCO-Host, Cochrane, ProQuest, SCOPUS and Google Scholar electronic databases, utilized specific keywords (“Probiotic” OR “Pro-biotics”) AND (“Colonoscopy” OR “Colonoscopies”).”
Comment 2: Re-analyze the included RCTs rather than briefly telling the results from each trial.
Response 2: As you rightly suggested, I focused on the 3 double-blind RCT studies by modifying the text between lines 97-106 as follows: “In particular, all three published double-blind RCT studies used strains of lactobacilli in combination with bifidobacterium and highlighted a statistically significant improvement of gastrointestinal symptoms after endoscopic examination compared to placebo. 6,7,8. Respectively in these three studies, a reduction in the number of days of pain (8), con-stipation (7), and bloating (6) was observed. The indication for colonoscopy does not seem to influence the response to treat-ment, but in two randomized controlled trial, the administration of probiotics results to be more efficacy in subgroup of patients with preexisting symptoms before endo-scopic examinations. 6,8”
Comment 3: Discussion would benefit from mechanistic insights into strain-specific effects.
Response 3: Compared to the original version, I have added this part ""Although the mechanisms of actions of these two strains have not yet been entirely comprehended, Deng et al (5) observed that probiotics containing primarily Lactobacillus and Bifidobacterium could mitigate the negative effect of O2 over intake of bowel preparation on anaerobic intestinal bacteria (decrease of Bacteroidetes and Firmicutes and increase of Proteobacteria), as they greatly increased the abundance of Bacteroidetes. Butyrate produced by Bacteroidetes plays in fact an important role in maintaining the intestinal health of the host, exerting immunity and antitumor effect” in the text between lines 124-130 to discuss the potential mechanisms related to the beneficial strains effects.
Thank you very much for taking the time to review this manuscript.
Best regards,
Dr Matteo Piciucchi MD, PhD
Reviewer 4 Report
Comments and Suggestions for Authors
Journal: Medicine
Article: communication
Title: Is It Advisable to Use Probiotics Routinely After Colonoscopy? A Rapid Comprehensive Review of Evidences
Overview: this communication describes the current studies focused on evaluating the outcome probiotic treatment on ameliorating colonic dysbiosis observed in patients treated with bowel cleaning preparations prior to colonoscopy intervention.
Highlights: this contribution provides presumable mechanisms upon which the probiotics may prevent and /or ameriolate the post-colonoscopy symptoms to improve the quality of life in patients underwent this intervention.
Major comments:
1- authors must state explicitly the originality of this communication (line 70-71or the pertinent place according to the authors)
2- author must stick to the format of “communitacions” according the guide for authors of the journal “Medical Sciences”
Minor points
Line 23 and 73: define please “RCT”
Line 30. Correct please as “The aim…”
Lines 49-50 (or the pertinent place): authors should include examples of cleaning preparations most frequently used prior colonoscopy intervention
Table 1 clarify please the term “high quality” and “low quality” included in some studies
Table 1 clarify which probiotic and placebo formulations were used for oral administration in most trials i.e. capsules? suspensions?,
Author Response
Dear Reviewer,
thank you for the time you dedicated to evaluating and correcting our article. Your comments were useful and valuable in improving the quality of the work. Below, I report your comments and my responses along with the corresponding changes in the text of the latest revised version of article.
Major comments:
Comment 1: authors must state explicitly the originality of this communication (line 70-71or the pertinent place according to the authors)
Response 1: In accordance with your valid observation, I have added this sentence from line 73 to line 76: “Since this topic has been little addressed in literature yet, the aim of this rapid communication review is to evaluate and summarized the available scientific evidence supporting the use of probiotics post colonoscopy, to define their real efficacy as a rou-tine treatment in this clinical setting.”
Comment 2- author must stick to the format of “communication” according the guide for authors of the journal “Medical Sciences”
Response 2: To better adhere to the journal's guidelines, I structured the article by dividing it into 4 sections: introduction and aim, methods, results and discussion. In this way, the article is aligned in length and structure with the “Medical Sciences” guidelines.
Minor points
Comment 3: Line 23 and 73: define please “RCT”
Response 3: The RCT abbreviation has been corrected with Randomized controlled trial
Comment 4: Line 30. Correct please as “The aim…”
Response 4: The grammatical error has been corrected
Comment 5: Lines 49-50 (or the pertinent place): authors should include examples of cleaning preparations most frequently used prior colonoscopy intervention
Response 5: Between lines 50-51 I have added the following sentence “(single or split dose, high or low volume preparation).”
Comment 6: clarify please the term “high quality” and “low quality” included in some studies.
Response 6: In the summary table, when the study was considered of low quality, the reason for attributing this status was specified in parentheses in the first column.
Comment 7: Table 1 clarify which probiotic and placebo formulations were used for oral administration in most trials i.e. capsules? suspensions?
Response 7: The title of the third column of the table has been changed to “Probiotics mixture and formulation” and formulation of probiotic and placebo (where assumed) have been specified.
Best regards,
Dr Matteo Piciucchi MD, PhD
Corresponding author
Round 2
Reviewer 2 Report
Comments and Suggestions for Authors
Thank you for the changes.
The benefit of probiotics in this setting would need futher exploration before any firm conclusions can be made.
Author Response
Dear reviewer,
I am the one who thanks you for the time spent evaluating my work. I completely agree with your assessment and in fact, in the conclusion of the article (lines 150-152), I have emphasized it this concept "further several prospective multi-arm case-control studies on large case series are certainly needed to establish the real efficacy and necessity of probiotic treatments after colonoscopy. "
I hope that the work in its current form meets your needs and you approve its publication.
Best regards,
Dr Matteo Piciucchi
Reviewer 3 Report
Comments and Suggestions for Authors
Thanks for the authors' effort to improve the manuscript. However, I think my previous concerns are not well addressed.
(1) besides clarifying the search strategy, use quality assessment tools to evaluate evidence reliability.
(2) apply quantitative meta-analysis rather than just description.
(3) clearly distinguish evidence-based conclusions from speculative hypotheses. It seems no direct evidence to conclude E. Coli nonpathogenic strains may constitute a valid alternative treatment.
Author Response
Dear reviewer,
I thank you for the time you dedicated to reassessing my work. I have tried as much as possible to follow your guidelines, while not distorting the simple nature of this brief narrative communication. I hope that these further modifications made can meet your requests and that you will approve the publication of the article.
Comment 1: besides clarifying the search strategy, use quality assessment tools to evaluate evidence reliability.
Response 1: In the first column of the table we assigned a quality rating (high, moderate, low) for each study based on the GRADE approach. For studies where a low to moderate value of quality was assigned, it was specified what limitations justified this rating.
Comment 2: Apply quantitative meta-analysis rather than just description.
Response 2: The type of article we have written is neither a meta-analysis nor a systematic review, but an invited article of type brief narrative communication of 2 pages. I believe that the type of more sophisticated analysis you are requesting does not fall within the scope of this simpler type of article. I hope you understand that the type of analysis you are requesting would completely change the profile and the approach of what would instead be an article (which, while as rigorous as possible) should be simple and easy to understand and read.
Comment 3: clearly distinguish evidence-based conclusions from speculative hypotheses. It seems no direct evidence to conclude E. Coli nonpathogenic strains may constitute a valid alternative treatment.
Response 3: Your consideration is indeed absolutely correct; therefore, I have modified lines 143-150 as follows to emphasize the speculative nature of the assumption: “On the basis of this hypothesis, it might seem conceivable that probiotic strains of O2-consumers extracted from the feces of healthy newborns could constitute a valid aid for a physiological bacterial recolonization of the colon after the intake of a purgative for colonoscopy. Therefore, in purely theoretical terms, we hypothesize that the administration of non-pathogenic neonatal strains of Enterobacteriacea such as the Escherichia Coli (which, as we previously mentioned, are markedly reduced in the long term after intake of the preparation) could constitute an adequate treatment for post colonoscopy syndrome, but clinical studies are need to confirm this supposition”
Best regards,
Dr Matteo Piciucchi MD, PhD
Round 3
Reviewer 2 Report
Comments and Suggestions for Authors
See before
Author Response
Dear reviewer,
I thank you for your previous comments. In this round, you did not provide specific feedback on my last revision, so I hope that the last revision I made has sufficiently met your requirements for publication. Thank you for the time you have dedicated to my work.
Best regards,
Dr Matteo Piciucchi MD, PhD
Reviewer 3 Report
Comments and Suggestions for Authors
issues addressed.
Author Response

(The authors gave the same response as above.)
